# A Taper-in-Taper Structured Interferometric Optical Fiber Sensor for Cu^2+^ ion Detection

**DOI:** 10.3390/s22072709

**Published:** 2022-04-01

**Authors:** Zidan Gong, Yisong Lei, Ziwen Wang, Jie Zhang, Zeji Sun, Yuyao Li, Jianhao Huang, Chichiu Chan, Xia Ouyang

**Affiliations:** 1Sino German College of Intelligent Manufacturing, Shenzhen Technology University, Shenzhen 518118, China; leiyisong2020@email.szu.edu.cn (Y.L.); 2070412024@email.szu.edu.cn (Z.W.); 2110412004@stumail.sztu.edu.cn (J.Z.); 201901010403@stumail.sztu.edu.cn (Z.S.); 201901010404@stumail.sztu.edu.cn (Y.L.); 201901010406@stumail.sztu.edu.cn (J.H.); xia.ouyang@foxmail.com (X.O.); 2Center for Smart Sensing System, Julong College, Shenzhen Technology University, Shenzhen 518118, China; chenzhichao@sztu.edu.cn; 3Department of Mechanical Engineering, University of Minnesota, Minneapolis, MN 55455, USA

**Keywords:** optical fiber sensor, interferometric, taper-in-taper, refractive index, Cu^2+^ detection

## Abstract

Copper ion is closely associated with the ecosystem and human health, and even a little excessive dose in drinking water may result in a range of health problems. However, it remains challenging to produce a highly sensitive, reliable, cost-effective and electromagnetic-interference interference-immune device to detect Cu^2+^ ion in drinking water. In this paper, a taper-in-taper fiber sensor was fabricated with high sensitivity by mode-mode interference and deposited polyelectrolyte layers for Cu^2+^ detection. We propose a new structure which forms a secondary taper in the middle of the single-mode fiber through two-arc discharge. Experimental results show that the newly developed fiber sensor possesses a sensitivity of 2741 nm/RIU in refractive index (RI), exhibits 3.7 times sensitivity enhancement when compared with traditional tapered fiber sensors. To apply this sensor in copper ions detection, the results present that when the concentration of Cu^2+^ is 0–0.1 mM, the sensitivity could reach 78.03 nm/mM. The taper-in-taper fiber sensor exhibits high sensitivity with good stability and mechanical strength which has great potential to be applied in the detection of low Cu^2+^ ions in some specific environments such as drinking water.

## 1. Introduction

Fiber optic sensors have drawn considerable interest because of the ability for monitoring the environmental variables such as temperature, stress, relative humidity, RI, etc. [1,2,3,4,5]. Such sensors have been widely applied in physical, chemical, and biological applications due to their unique properties (e.g., micro-size, lightweight, flexible, durable, biocompatibility, corrosion resistance, cost-effective, immune to electromagnetic interference) [6,7,8,9]. To date, several fiber-optic sensors with different structures have been developed, including photonic crystal fibers [10,11], FBGs [12,13], long-period fiber gratings [14,15], optical microfibers [16], and tapered optical fibers [17,18].

Recently, tapered optical fibers have become one of the current trends among optical sensors given the rapid progress in tapered optical fibers and the great demands on optical sensors with higher sensitivity and simpler structure. Based on Mach–Zehnder interferometers (MZIs) with strong evanescent field, fiber sensors with tapered structures can realize higher sensitivity compared with common and mature fiber structures such as fiber Bragg grating sensors [19,20]. Among a variety of tapered fiber sensors based on MZI, RI sensors have been intensively studied because RI itself is an important parameter in physical and chemical fields that could realize the indirect measurement of many other parameters such as solution concentration and pH value [21,22,23]. MZI-based RI sensor can split incident light into two arms and recombine them via a second coupler with a unique compact structure, which presenting high sensitivity in testing environment when surrounded by mediums and showing good characterizes in facile setup and direct readout [24,25,26]. Wu et al. [27] proposed an MZI-based RI sensor by splicing hollow quartz tube with single mode fiber in tapered shape, which achieved a sensitivity of −120.18 dB/RIU, but this structure requires complex splicing technique and the sensitivity of which was still not yet high enough. Ma et al. [28] proposed an S-shaped tapered fiber RI sensor with a sensitivity of 2109.7 nm/RIU that need to be fabricated via precise controlling of the axial offset, otherwise the sensing performance would be affected. Thereby, many MZI-based RI sensors require complex fabrication techniques and lack reproducibility and test stability that is worth the research effort to make improvements. Currently, a taper-in-taper structured fiber optic sensor was put forward with enhanced sensitivity, excellent linear relationship and simple fabrication process which could be flexibly applied in different fields [29,30].

With the rapid development of industrialization, the heavy metal pollution in soil, air and water has become a serious problem, one which is highly associated with the ecosystem and human healthcare [31,32]. As one of the heavy metals, Cu^2+^, even a low concentration in drinking water, would result in a number of health problems, such as kidney diseases, Wilson’s disease, Alzheimer’s disease and prion diseases [33,34,35,36]. The maximum Cu^2+^ concentration in drinking water is limit to 20 μM, set by the U.S. Environmental Protection Agency [35]. Thereby, copper ions detecting is necessary which plays an important role in the management of water resource and healthcare. Traditional methods for detecting metal ions including Cu^2+^ involved atomic absorption spectrometry (AAS) [37], atomic fluorescence spectrometry (AFS) [38], spectrophotometry [39], electrochemical analysis [40] and inductively coupled plasma spectrometry [41]. However, these methods require high investment, complex sample pretreatment, and not sensitive enough for low concentration detection. Thus, optical fiber sensing technology could be potentially applied in Cu^2+^ detection due to its aforementioned unique advantages. Sung et al. [42] proposed using a high sensitive fiber sensor based on CdSe/ZnS QD for Cu^2+^ ion detection via fluorescence quenching, but there is a high temperature cross interference on fluorescence intensity variation. Huang et al. [43] developed an ultra-sensitive optical fiber plasmonic sensor for Cu^2+^ detection with a multimode-single-mode-multimode structure which requires complicated fabrication techniques and is also costly. Additionally, Tang et al. [44] proposed a long-period grating fiber sensor for the detection of Cu^2+^ with the maximum sensitivity of 26.1265 nm/mM, which is not high enough in practical use. Moreover, few previous studies adopted tapered fiber for the detection of Cu^2+^.

Here, we developed a taper-in-taper structured fiber sensor with high sensitivity by mode-mode interference, and systematically explore the influence of physical parameters of fiber on sensing performance to make optimization. The proposed interferometric sensor presents a high sensitivity of 2741 nm/RIU between RI 1.38027 and 1.40169, and the sensitivity of 1552 nm/RIU within the RI range from 1.33300 to 1.38027. To apply in Cu^2+^ detection, the chitosan/polyacrylic acid (PAA) polyelectrolyte layer was coated on the fiber surface. The sensitivity could reach 78.03 nm/mM when the Cu^2+^ concentration is from 0 to 0.1 mM, while within the concentration of 0.1–0.7 mM, the sensitivity is 14.59 nm/mM. Thus, the developed taper-in-taper structure created a sensitivity-enhanced interferometric sensor, which could be potentially apply in Cu^2+^ detection especially among low concentration conditions.

## 2. Materials and Methods

### 2.1. Fabrication of the Taper-in-Taper Fiber Sensor

The commercially available G.625.D single-mode fiber was adopted to fabricate the tapered optical sensor as schematically shown in Figure 1a. Unlike traditional fabrication method of tapered fibers in most previous studies, the Large Diameter Splicing System (LDS 2.5, 3SAE Technologies, Inc., Franklin, TN, USA) was used instead of oxyhydrogen flame to obtain more accurate taper with the minimum waist diameter at the micrometer level. To fabricate the sensing structure, fiber with the coating layer stripped was placed on the holder of the LDS near the electrode as presented in Figure 1b. The LDS works by discharging to the fiber via electric arc and applying constant tension to one end of the fiber to achieve the tapered structure. The sweep speed of the electric arc, start speed, and start power were set at 600 μm/s, 50 μm/s, and 600 (unit), respectively, on the control panel while the whole tapering process only lasts about 45 s. Programs and with predefined parameters could be saved in LDS for repetitive fabrication. To obtain a symmetrical structure on both ends of the tapered fiber, fiber on holders should not be fixed too tight as it may cause excessive deformation. In this study, a series of taper-in-taper fiber sensors with different diameters, lengths and taper ratios were manufactured and studied systematically for optimization.

### 2.2. RI and Cu^2+^ Detection Experiments

In the RI detection experiment, one end of the developed sensor was spliced with super broad-spectrum light source (SBS, SC-5, Wuhan Anyang Laser Technology Co., Ltd., Wuhan, China), and the other end was connected to optical spectrum analyzer (OSA, AQ6370D, Yokogawa Ltd., Tokyo, Japan) to record the optical spectrum variation. The sensing structure was fixed on a pair of fiber holders (xyz60, Beijing Optical Century Instrument Co., Ltd., Beijing, China). A glass slide for carrying liquids was placed on the manual lifting platform under the fiber, which could be adjusted to emerge sensing area into the liquid. The experiment set up is presented in Figure 2a. Thirteen types of liquids with RI ranging from 1.33300 to 1.40169 were prepared by matching the glycerol into deionized water at a certain percentage; nine Cu^2+^ solutions with different concentrations range from 0–1 mM were prepared by diluting copper chloride dihydrate (CuCl_2_·2H_2_O). Once the constant spectrum data of one liquid were recorded, the sensing area need to be cleaned by deionized water before testing the next one. The whole experiments were conducted at 25 °C.

Chitosan is a natural polymer material derived from chitin, and is adopted in this study as the coating material due to its characteristics of non-toxic, tasteless, alkali resistance, corrosion resistance, biocompatibility and self-assembly capability [45]. Before the Cu^2+^ detection experiment, the developed sensor was functionalized by coating chitosan/Polyacrylic acid film as shown in Figure 2b that could chelate with Cu^2+^ resulting in RI variation. Specifically, the sensor was immersed in piranha solution (H_2_SO_4_:H_2_O_2_ = 7:3 (*v*/*v*)) for 1 h to achieve surface hydroxylation. Then, 0.4 g chitosan was dissolved in 50 mL 4% acetic acid and stirred at room temperature (25 °C) for 24 h to obtain the chitosan solution. An amount of 10% PAA solution was prepared with deionized water. The chitosan solution was titrated to the sensor and stored for 2 min before cleaning the excess chitosan solution by deionized water, then the sensor was dried for 2 min. The PAA solution was also dropped onto fiber and wash the excess PAA by deionized water then dried for 2 min. The alternate process was repeated 3 times to obtain the polyelectrolyte film structure. Cu^2+^ solutions with different concentrations were then tested in the similar experimental setting to the RI detection.

### 2.3. Principle of the Developed Sensor

During light propagating through the taper-in-taper structure, multiple high-order modes with different propagation constant are excited at the first and second taper, where part of light would leak out to generate the evanescent field. Then, these modes cause mode-mode interference and leaked light recoupled back into the fiber resulting in specific pattern spectrum for sensing application. The axial propagation constant difference between *m* and *n* order core modes can be expressed as [46]
(1)$$\beta_{m}-\beta_{n}=\frac{\mu_{m}^{2}-\mu_{n}^{2}}{2ka^{2}n_{core}},$$
where a is the core radius of the taper waist, *n_core_* is the core effective refractive index of the waist, *k* is the wave number, *µ_m_* and *µ_n_* are normalized transverse propagation constants given by *µ_x_* = (2*x* − 1/2) *π*/2. Constructive interference occurs when
(2)$$(\beta_{m}-\beta_{n})L=2\pi N,$$
where *L* is the length of waist, *N* is an integer, and the resonant wavelength can be derived from (1) and (2):(3)$$\lambda =\frac{8(2N+1)n_{core}a^{2}}{(m-n)[2(m+n)-1]L},$$

The wavelength difference of adjacent extreme is:(4)$$\Delta \lambda =\lambda_{N}-\lambda_{N-1}=\frac{16n_{core}a^{2}}{(m-n)[2(m+n)]L},\;$$

It can be derived from Equation (4) that Δ*λ* decreases when there is an increase in *L* and decrease in a. When the environmental RI changes, the propagation constants and mode field are reformed due to the change of the boundary condition for the light propagating in taper. Concluded from Equation (2), when the propagation constants are changed, the phase condition of the constructive (or destructive) interference is altered and *N* is changed. Thereby, wavelength of the constructive (or destructive) interference shifts which correspond to the surrounding RI value [47].

## 3. Results and Discussion

### 3.1. Sensitivity Study of Taper-in-Taper Fiber Sensor with Different Parameters

A series of taper-in-taper fiber sensors with various parameter design were produced as presented in Table 1. for sensitivity analysis. A controlled trial was conducted to systematically explore the influence of the waist diameter of the first and second tapers of the developed sensors on sensitivity performance.

Figure 3a illustrates that the sensor with the first taper waist diameter of 40 μm exhibits a relatively better sensitivity performance in both low and high RI liquids while other parameters being controlled. For the second taper waist, which is the main sensing area, a smaller diameter presents better sensitivity, as shown in Figure 3b, which is consistent with the aforementioned theory. However, a smaller diameter also means it is more fragile in practical use. Considering both mechanical strength and sensitivity, the second taper waist diameter of 20 μm was preferred under a controlled condition.

To further explore the structure optimization of the developed sensor, another controlled trial studying the influence of the second taper ratio (taper length/waist length) on RI sensitivity performance was conducted. Five sensors with different second taper ratios were fabricated as shown in Table 2.

It is clear in Figure 4 that along with the increase in the ratio of second taper, sensitivity performance of sensors would decrease in both low and high RI liquids, which is consistent with the results explored by the study of Wang et al. towards tapered fiber sensor [48]. A possible reason is that the decrease in taper ratio may change the launching angle inside the optical fiber as well as the penetration depth of the evanescent wave, thereby enhancing the sensitivity.

### 3.2. RI Testing Result of the Taper-in-Taper Fiber Sensor

Following the aforementioned experimental steps, the optimized taper-in-taper fiber sensor was fabricated with the microscope view presented in Figure 5a. The first and second tapers lengths are 800 μm and 600 μm, respectively; the waist diameter of the former is 40 μm, whereas that of the latter is 20 μm; the waist lengths of the first and second tapers are 900 and 2000 μm, respectively. The transmission spectrum of this developed sensor and its fast Fourier transform (FFT) were shown in Figure 5b,c. It can be clearly seen that the fundamental core mode energy near 0 is the highest and there are obvious peaks at 0.0479 nm^−1^, 0.0955 nm^−1^ and 0.1756 nm^−1^ due to the evanescent wave generated after tapering, which are the dominating cladding modes. Additionally, multiple higher-order cladding modes are also inspired, which enhance the response capability of the fiber sensor to environmental variables and improves the RI sensitivity.

The RI testing result of the optimized taper-in-taper fiber sensor is shown in Figure 6. With regard to the variation of the transmission optical spectrum of the sensor in 13 different RI liquids, the redshift occurs towards the interference fringe peaks and valleys along with the increase in liquid RI. A sensitivity of 1552 nm/RIU was obtained when the prepared liquids RI varies from 1.33300 to 1.38027, and even reach as high as 2741 nm/RIU when the liquid RI between 1.38027 and 1.40169. Obviously, an abrupt change in sensitivity occurred around RI 1.38. This may because the penetration depth of the evanescent wave would increase along with the external refractive index, thus more evanescent waves can break through the cladding and propagate forward along the surface of the cladding in the external medium environment. Therefore, when the external environment changes, variations in the evanescent wave would occur correspondingly, so as to realize the detection of external parameters with enhanced sensitivity, which has been previously reported then proved [49,50].

To better study the sensitivity performance of the newly developed taper-in-taper fiber sensor, two traditional tapered sensors (one with a waist diameter of 40 μm, waist length of 2000 μm, and taper length of 2300 μm; another with a waist diameter of 20 μm, waist length of 2000 μm, and taper length of 2300 μm) were prepared and tested for comparison. The waist length, tapered area length, and total length of the three sensors were exactly controlled to be the same. Testing results presented in Figure 7a indicates that for the traditional tapered fiber sensor whose waist diameter is 40 μm, although redshift occurred along with RI increased in liquids, a relatively low sensitivity of 379 nm/RIU was obtained within 1.33300 to 1.37073 RI liquid, and sensitivity of 741 nm/RIU was achieved in liquid RI from 1.37073 to 1.40169. For another traditional tapered sensor, whose diameter is 20 μm, a sensitivity of 877 nm/RIU was obtained among liquid RI from 1.33300 to 1.37073 and 2010 nm/RIU between 1.37073 and 1.40169, as illustrated in Figure 7b. By comparison, the developed taper-in-taper fiber sensor presented a significantly higher RI sensitivity.

To further verify the superiority of our developed structure, a taper-in-taper fiber sensor with the second taper waist diameter of 25 μm, and a single taper structure with the waist diameter of 25 μm were fabricated for comparation while all other parameters controlled to be the same. The result in Figure 8a indicates that the RI sensitivity of the taper-in-taper fiber sensor was 935 nm/RIU among the liquids RI from 1.33300 to 1.37073, and then reached 2268 nm/RIU when the liquid RI was between 1.37073 and 1.40169. Meanwhile, sensitivity of the single tapered fiber sensor is 439 nm/RIU in the RI range from 1.33300 to 1.37073, and 899 nm/RIU when liquid RI between 1.37073 and 1.40169 as shown in Figure 8b. Obviously, the taper-in-taper fiber sensor was significantly more sensitive to RI than traditional tapered sensor of the same physical parameters.

The excellent RI sensitivity performance may be due to the introduction of an evanescent wave caused by the taper-in-taper structure. When light passes through the first taper, multiple high-order modes with a different propagation constant are excited, and more high-order modes would be inspired while propagating through the second taper. Afterwards, mode-mode interference would occur among these modes resulting in specific pattern spectrum for sensing application. Similar structures such as cascade tapered sensor [51] and two-micro-bending-core sensor [52] have been proved to obtain applicable RI sensitivity that demonstrated the feasibility of our theory.

Moreover, the cross-sensitivity of temperature was considered in current study when applying this sensor in ambient RI measurement with the results presented in Figure 9. The taper-in-taper fiber sensor with the diameter of 20 μm in the second taper waist and a total length of about 6600 μm was put into deionized water heating from 25 °C to 70 °C. An interval of 5 °C was set up for the wavelength shift recording and the temperature sensitivity was tested to be 0.13935 nm/°C. As a result, when comparing with the significantly enhanced RI sensitivity, the temperature cross interference could be ignored.

Table 3 lists other reported MZI-based fiber optic sensors. Compared with the current study, our taper-in-taper sensor has presented a higher sensitivity as well as a higher measurement range.

### 3.3. Application in Cu^2+^ Detection

To apply this optimized taper-in-taper fiber sensor (Sensor No.: a-2) in Cu^2+^ detection, the fiber surface was functionalized and coated with a chitosan/PAA molecular film, then immersed into Cu^2+^ solutions with different concentrations. The results in Figure 10a observed a wavelength shift as the concentration of Cu^2+^ increases, because of the chelation between Cu^2+^ and the sensing film. Linear fitting outcomes in Figure 10b indicate a sensitivity as high as 78.03 nm/mM in external Cu^2+^ concentration from 0 to 0.1 mM. When the concentration increased to the range from 0.1 to 0.7 mM, a decreased sensitivity of 14.59 nm/mM was obtained due to saturation of the film resulting in a weakened chelation. Afterwards, the spectrum gradually becomes stable. Therefore, this developed taper-in-taper fiber sensor could have great potential to be applied in low concentration Cu^2+^ detection.

In addition, the functionalized optical fiber sensor for Cu^2+^ detection was tested for the cross-sensitivity of temperature with the results illustrated in Figure 11. A temperature sensitivity of 0.14 nm/°C was obtained by the sensor, which is close to that of the unfunctionalized one, indicating that the proposed taper-in-taper sensor meet the daily detection requirements.

## 4. Conclusions

This study demonstrated a newly developed interferometric sensor for RI and Cu^2+^ detection. A novel taper-in-taper structure was designed by tapering on a tapered fiber sensor, wherein the diameter of the first taper waist is twice that of the second one. The RI sensitivity of the sensor was obtained as high as 2741 nm/RIU when the liquids RI from 1.38027 to 1.40169, surpassing a series of traditional tapered fiber sensors. Additionally, it has been applied in the detection of Cu^2+^ among various concentrations. When the concentration of Cu^2+^ is 0–0.1 mM, the sensor sensitivity is as high as 78.03 nm/mM, and when the concentration is 0.1–0.7 mM, the sensitivity is 14.59 nm/mM. To conclude, the taper-in-taper fiber sensor exhibits significantly high sensitivity, with other advances of low cost, good test stability and mechanical strength, which indicates great potential to be applied in low concentration Cu^2+^ detection in some specific environments such as drinking water.

## Figures and Tables

**Figure 1 sensors-22-02709-f001:**
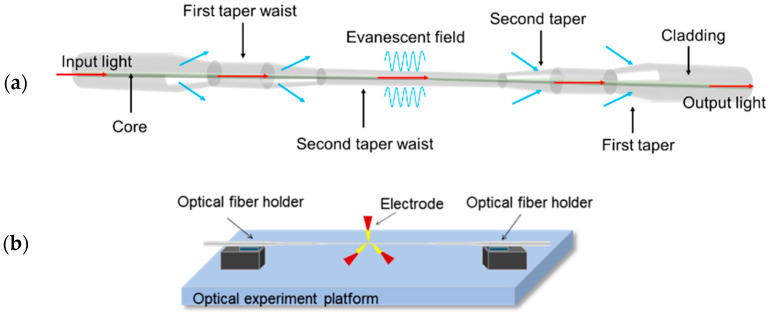
(**a**) Schematic structure of the taper-in-taper fiber sensor; (**b**) The tapering process.

**Figure 2 sensors-22-02709-f002:**
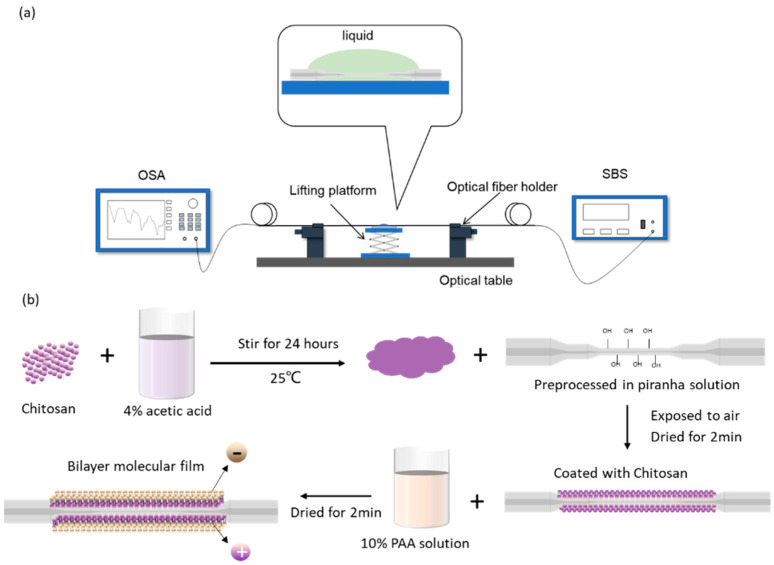
(**a**) The experiment setup; (**b**) Chitosan/PAA polyelectrolyte film coating process.

**Figure 3 sensors-22-02709-f003:**
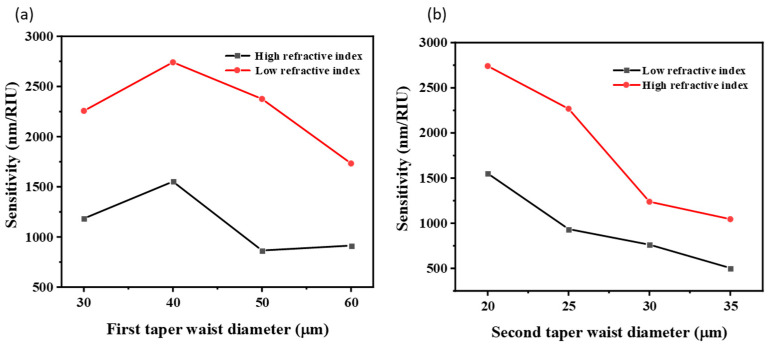
The influence of (**a**) the first taper waist diameter and (**b**) the second waist diameter on sensitivity performance of the taper-in-taper fiber sensor.

**Figure 4 sensors-22-02709-f004:**
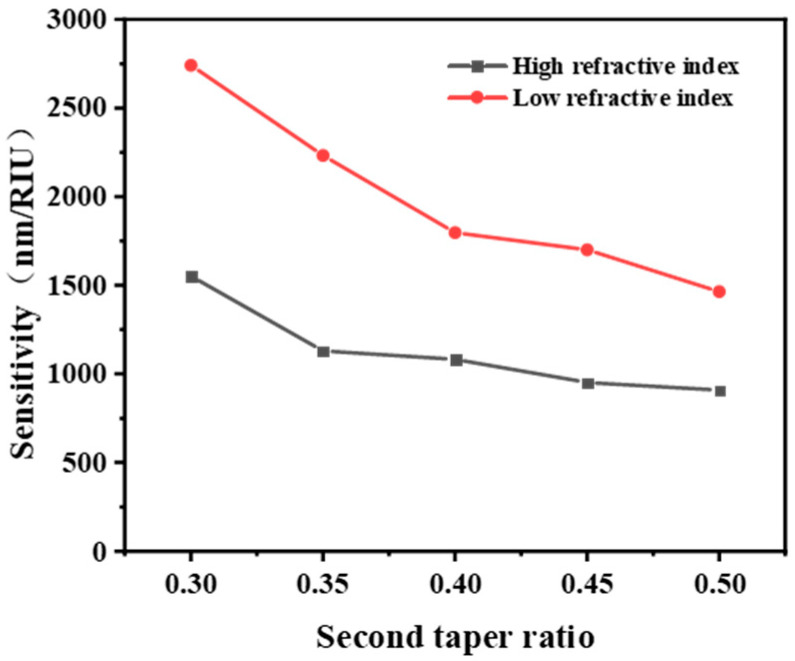
Influence of the second taper ratio on sensitivity performance of the taper-in-taper fiber sensor.

**Figure 5 sensors-22-02709-f005:**
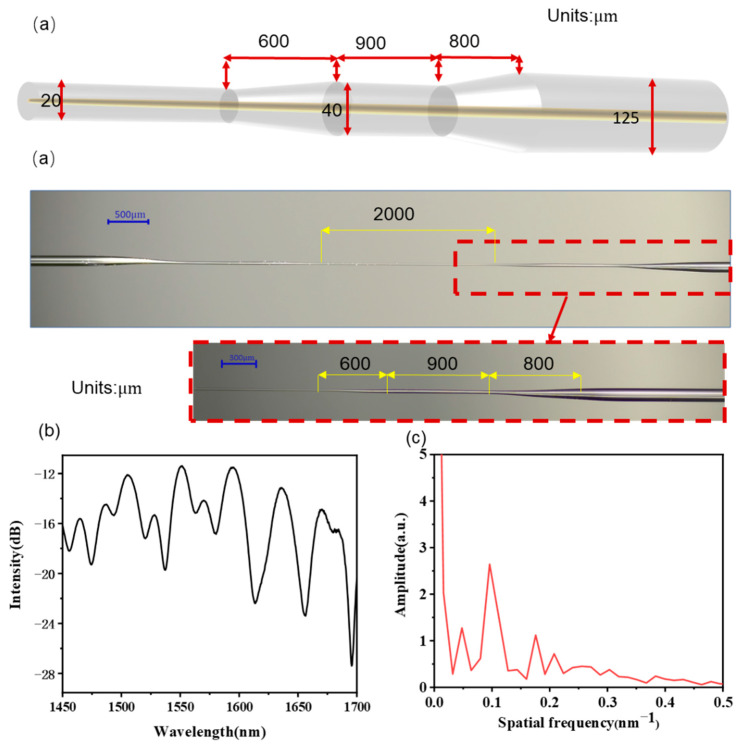
(**a**) Microscope view of the developed sensor; (**b**) Transmission spectra and its (**c**) FFT spectrum.

**Figure 6 sensors-22-02709-f006:**
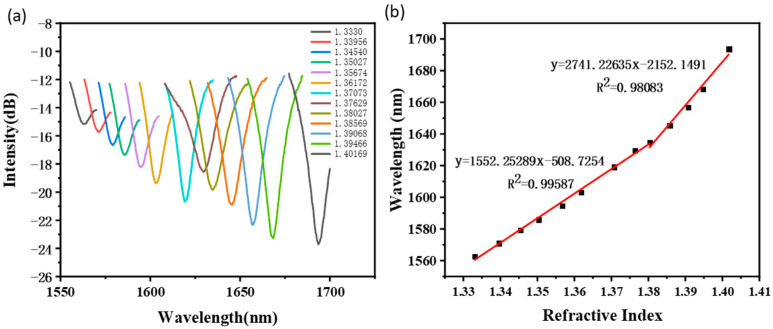
RI testing result of the taper-in-taper fiber sensor: (**a**) Wavelength shift; (**b**) Linear fitting result.

**Figure 7 sensors-22-02709-f007:**
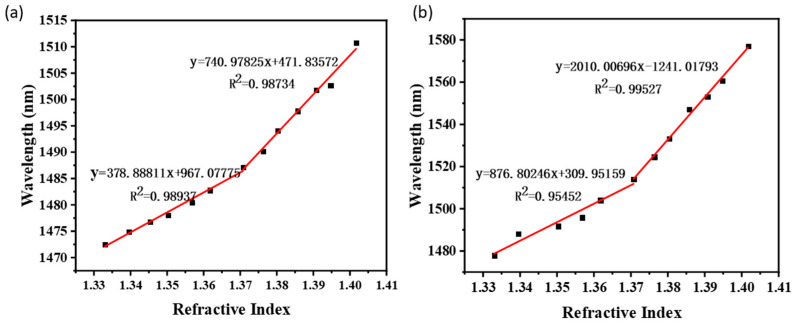
Linear fitting results of two traditional tapered sensors: (**a**) sensor with the waist diameter of a 40 μm and (**b**) sensor with the waist diameter of 20 μm.

**Figure 8 sensors-22-02709-f008:**
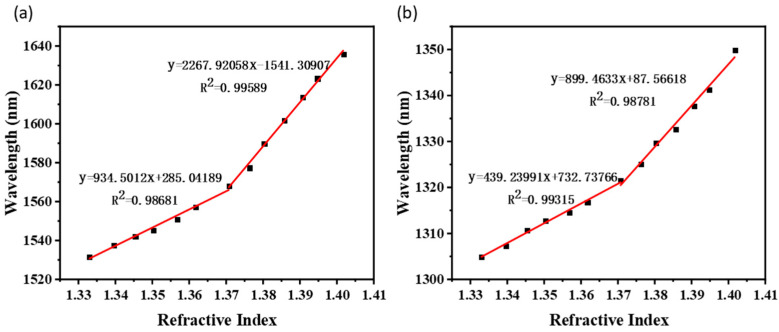
Linear fitting result of (**a**) the tape-in-taper fiber sensor with the second taper waist diameter of 25 μm and (**b**) the single taper fiber sensor with the taper waist diameter of 25 μm.

**Figure 9 sensors-22-02709-f009:**
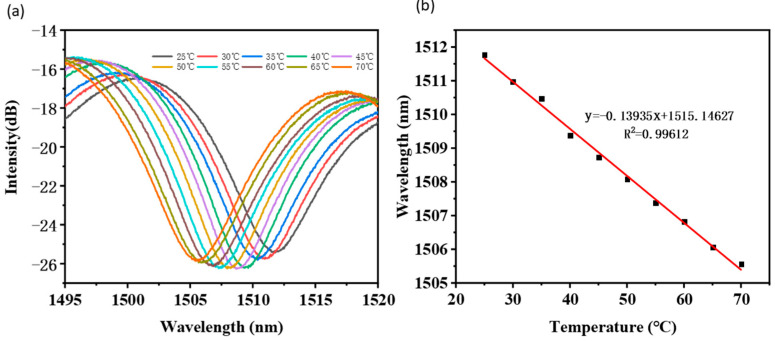
Testing on cross-sensitivity of temperature: (**a**) Wavelength shift and (**b**) its linear fitting result.

**Figure 10 sensors-22-02709-f010:**
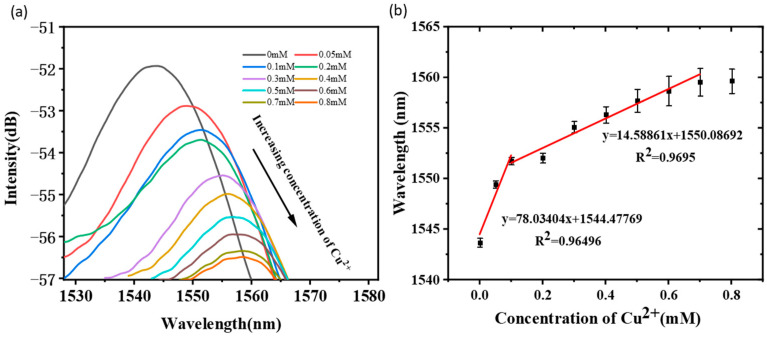
Detection results in various Cu^2+^ concentrations: (**a**) Wavelength shift and (**b**) its linear fitting result.

**Figure 11 sensors-22-02709-f011:**
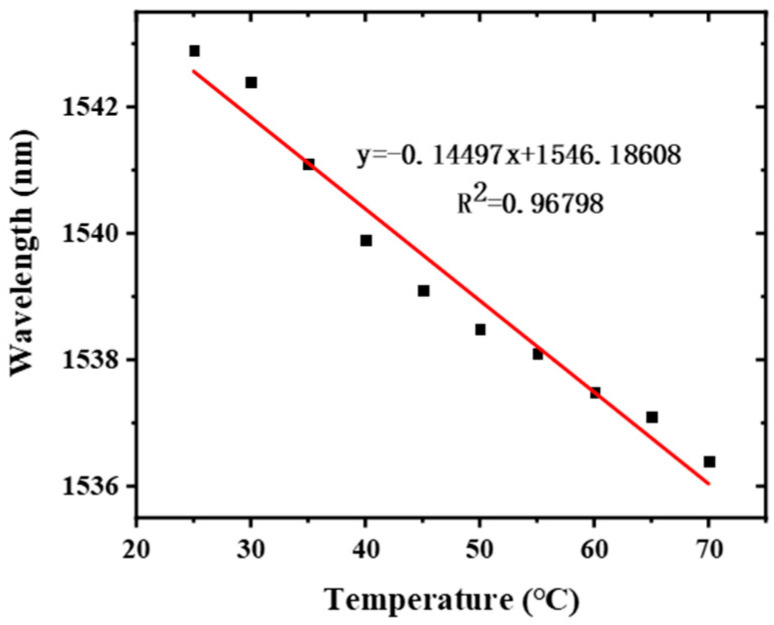
Temperature sensitivity of the functionalized fiber optic sensor.

**Table 1 sensors-22-02709-t001:** Tapered fiber sensors with various diameters of first and second taper waist.

Sensor No.	Diameter of the First Taper Waist (μm)	Diameter of the Second Taper Waist (μm)	Second Taper Length (μm)	Second Taper Waist Length (μm)	Second Taper Ratio	Sensor Total Length (μm)
a-1	30	20	600	2000	0.3	6600
a-2	40	20	600	2000	0.3	6600
a-3	50	20	600	2000	0.3	6600
a-4	60	20	600	2000	0.3	6600
b-1	40	20	600	2000	0.3	6600
b-2	40	25	600	2000	0.3	6600
b-3	40	30	600	2000	0.3	6600
b-4	40	35	600	2000	0.3	6600

**Table 2 sensors-22-02709-t002:** Tapered fiber sensors with different second taper ratios.

Sensor No.	Diameter of the First Taper Waist (μm)	Diameter of the Second Taper Waist (μm)	Second Taper Length (μm)	Second Taper Waist Length (μm)	Second Taper Ratio	Sensor Length (μm)
c-1	40	20	600	2000	0.3	6600
c-2	40	20	700	2000	0.35	6700
c-3	40	20	800	2000	0.4	6800
c-4	40	20	900	2000	0.45	6900
c-5	40	20	1000	2000	0.5	7000

**Table 3 sensors-22-02709-t003:** Comparison with other published wavelength-modulated RI sensors.

Refs.	Sensing Principle	RI Sensitivity (RIU)	Measurement RANGE (nm/RIU)
[53]	MZI-taper	1.333–1.337	1905.7
[28]	MZI-taper	1.360–1.385	2109.7
[52]	MZI	1.33–1.42	−333.8
[54]	MZI-taper	1.36–1.42	108.07
Our work	MZI-taper	1.33300–1.40169	2471

## Data Availability

The data presented in this study are available on request from the corresponding author. The data are not publicly available due to privacy issues.

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
