# Peer review of "A Taper-in-Taper Structured Interferometric Optical Fiber Sensor for Cu2+ ion Detection"

_sensors, 2022, doi:10.3390/s22072709_

Round 1

Reviewer 1 Report

Fabrication process
The fabrication process of taper in taper fiber is not clearly explained. Please explain the process of generating the two steps of tapers for readers. And what parameters were changed to achieve the diameters, lengths, and taper ratios to optimize the structures? This information will be helpful for readers to understand how specific parameter affects diameters, taper length, and taper ratio.

Principle of the sensor
Please elaborate on how multiple higher-order cladding modes can enhance the proposed sensor's response to environmental variables. In lines 227 - 229, it is said that higher sensitivity is achieved by increasing the portion of the evanescent wave. If the mode is guided through the cladding, the interaction between the mode and the environment should be limited. More explanation will be helpful for casual readers on this matter.

Temperature dependency of the sensor
In order to ignore the temperature sensitivity, the peak shift resulting from the nominal temperature change and the peak shift resulting from nominal copper ion detection should be compared. Please provide a realistic comparison of those two peak shifts.

Miscellaneous
Line 107: taping -> tapering

Author Response

Pleanse see the attachment. 

Reviewer 2 Report

This paper reports a taper-in-taper structured interferometric optical fiber sensor for Cu2+ ion detection. I have questions:

  1. Remove "novel" words from title. there is a lot literature using taper-in-taper structure and was ignored from a rapid checking in literature. Please read and mention: Optics Express 29 (26), 43793-43810, 2021; IEEE Transactions on NanoBioscience 20 (3), 377-384, 2021. Both literature were ignored and need to be discussed and show the novelty of your structure.
  2. How is the reproducibility to get identical probes?
  3. How about the complexity for handling such tapers and how you can control the stability spectrum?
  4. Fig. 10: how many probes were tested? Nothing about error bars.
  5.  Fig. 5b) at least three spectra from 3 different probes are needed to show to see the differences.

Round 2

Reviewer 2 Report

The authors need to improve the introduction with the literature suggested since it was again ignored.

Author Response

Dear Editor and Reviewer

Thanks for your precious and constructive comments on the manuscript again. Literature suggested have been added to the introduction for improvement highlighted in Line 41-44, 58-61.

Yours sincerely,

Gong Zidan (Dr.)

Sino-German College of Intelligent Manufacturing

Shen Zhen Technology University

3002 Lantian Road, Pingshan District, Shenzhen Guangdong, China, 518118

Tel: 0755-23256330

E-mail: gongzidan@sztu.edu.cn

Round 3

Reviewer 2 Report

It is ok for publication